# HyperTime: Implicit Neural Representations for Time Series Generation

## Abstract

Implicit neural representations (INRs) have recently emerged as a powerful tool that provides an accurate and resolution-independent encoding of data. Their robustness as general approximators has been shown in a wide variety of data sources, with applications on image, sound, and 3D scene representation. However, little attention has been given to leveraging these architectures for the representation and analysis of time series data. In this paper, we propose a new INR architecture for time series (iSIREN) designed to perform an accurate reconstruction of univariate and multivariate data, while also providing an interpretable encoding of the signal. We compare our architecture against SIREN and INRs with different activations, in terms of training convergence, and the reconstruction accuracy of both the signal and its spectral distribution. To achieve generalization, we propose a hypernetwork architecture (HyperTime) that leverages iSIRENs to learn a latent representation of an entire time series dataset. In addition to the traditional reconstruction loss, we introduce an FFT-based loss that guides the training by enforcing a good match of the ground truth spectral distribution. We show how these architectures can be used for time series generation, and evaluate our method through fidelity metrics, presenting results that exceed the performance of state-of-the-art techniques. Finally, we propose an alternative hypernetwork architecture (iHyperTime) that incorporates interpretability into the latent representation, enabling the introduction of prior knowledge by imposing constraints into the generation process.

## 1 Introduction

Modeling time series data has been a key topic of research for many years, constituting a crucial component in a wide variety of areas such as climate modeling, medicine, biology, retail and finance (Lim & Zohren, 2021). Traditional methods for time series modeling have relied on parametric models informed by expert knowledge. However, the development of modern machine learning methods has provided purely data-driven techniques to learn temporal relationships. In particular, neural network-based methods have gained popularity in recent times, with applications to a wide range of tasks, such as time series classification (Ismail Fawaz et al., 2020), clustering (Ma et al., 2019; Alqahtani et al., 2021), segmentation (Perslev et al., 2019; Zeng et al., 2022), anomaly detection (Choi et al., 2021; Xu et al., 2018; Hundman et al., 2018), upsampling (Oh et al., 2020; Bellos et al., 2019), imputation (Liu, 2018; Luo et al., 2018; Cao et al., 2018), forecasting (Lim & Zohren, 2021; Torres et al., 2021) and synthesis (Alaa et al., 2021; Yoon et al., 2019b; Jordon et al., 2019). In particular, generation of synthetic time series has recently gained attention due to the large number of potential applications in medical and financial fields, where data cannot be shared, either due to privacy reasons or proprietary restrictions (Jordon et al., 2021; 2019; Assefa et al., 2020). Moreover, synthetic time series can be used to augment training datasets to improve model generalization on downstream tasks, such as classification (Fons et al., 2021), forecasting and anomaly detection.

In recent years, implicit neural representations (INRs) have gained popularity as an accurate and flexible method to parameterize signals from diverse sources, such as images, video, audio and 3D scene data (Sitzmann et al., 2020b; Mildenhall et al., 2020). Conventional methods for data encoding often rely on discrete representations, such as data grids, which are limited by their spatial resolution and present inherent discretization artifacts. In contrast, INRs encode data in terms of continuous functional relationships between signals, and thus are uncoupled to spatial resolution. In practical terms, INRs provide a new data representation framework that is resolution-independent,

with many potential applications to time series, where missing values and irregularly sampled data are common occurrences (Fang & Wang, 2020). While there have been a few recent works exploring the application of INRs to time series data, there is virtually no work on leveraging these architectures for generating synthetic time series or producing interpretable time series representations (Jeong & Shin, 2022; Woo et al., 2022). Applications with regulatory focus such as finance often require transparency and interpretability of proposed machine learning solutions as well as injection of expert knowledge as constraints into the training process to guide learning. For instance, explainable construction of trading agents is preferred when actions of trading agents need to be explicitly attributed to market signals (Vyetrenko & Xu, 2019).

In this paper, we propose a novel methodology that utilizes INRs to encode and generate time series data based on interpretable latent representations. To the best of our knowledge, we are the first to incorporate an interpretable decomposition into the generation of time-series. Our contributions are as follows:

**Representation and Generation of time-series using INRs:** We introduce iSIREN, an INR architecture for multivariate time-series representation which provides an interpretable trend-seasonality decomposition of the data. We show that interpretability does not lead to a loss of reconstruction accuracy, and in some cases increases the spectral reconstruction quality. Moreover, we leverage a hypernetwork for time-series generation via interpolation of learned embeddings.

**Spectral Loss:** To improve the training of the hypernetwork, we introduce a novel spectral loss that enforces the correct reconstruction of the signal's spectral distribution. We show that for some datasets this loss plays a crucial role in the learning process.

**Interpretable time-series generation:** We propose iHyperTime, a hypernetwork architecture for time-series generation that learns a disentangled seasonal-trend representation of time series, enabling the introduction of expert knowledge into the synthesis process. We compare iHyperTime against current state-of-the-art methods for time-series generation, showing improved results in terms of standard fidelity metrics.

## 2 RELATED WORK

**Implicit Neural Representations** Implicit Neural Representations (INRs) provide a continuous representation of multidimensional data, by encoding a functional relationship between input co-ordinates and signal values, avoiding possible discretization artifacts. They have recently gained popularity in visual computing (Mescheder et al., 2019; Mildenhall et al., 2020) due to the key development of positional encodings (Tancik et al., 2020) and periodic activations (SIREN (Sitzmann et al., 2020b)), which have proven to be critical for the learning of high-frequency details. Whilst INRs have been shown to produce accurate reconstructions in a wide variety of data sources, such as video, images and audio (Sitzmann et al., 2020b; Chen et al., 2021; Rott Shaham et al., 2021), few works have leveraged them for time series representation (Jeong & Shin, 2022; Woo et al., 2022), and none have focused on interpretability and generation.

**Hypernetworks** Hypernetworks are neural network architectures that are trained to predict the parameters of secondary networks, referred to as Hyponetworks (Ha et al., 2017; Sitzmann et al., 2020a). In the last few years, some works have leveraged different hypernetwork architectures for the prediction of INR weights, in order to learn priors over image data (Skorokhodov et al., 2021) and 3D scene data (Littwin & Wolf, 2019; Sitzmann et al., 2019; Sztrajman et al., 2021). Sitzmann et al. (2020b) leverage a set encoder and a hypernetwork decoder to learn a prior over SIRENs encoding image data, and apply it for image in-painting. Our HyperTime and iHyperTime architectures detailed in Section 3 use a similar encoder-decoder structure, however they learn a latent representation over our interpretable SIREN INRs (iSIREN), which encode time series data. Moreover, we apply these architectures for time series generation via interpolation of learned embeddings.

**Time Series Generation** Synthesis of time series data using deep generative models has been previously studied in the literature. Examples include the TimeGAN architecture (Yoon et al., 2019a), as well as QuantGAN (Wiese et al., 2020). More recently, Desai et al. (2021) proposed TimeVAE as a variational autoencoder alternative to GAN-based time series generation. Alaa et al. (2021) introduced Fourier Flows, a normalizing flow model for time series data that leverages the frequency domain representation, which is currently considered together with TimeGAN as state-

of-the-art for time series generation. In the last few years, multiple methods have used INRs for data generation, with applications on image synthesis (Chan et al., 2021; Skorokhodov et al., 2021), super-resolution (Chen et al., 2021) and panorama synthesis (Anokhin et al., 2021). However, there are currently no applications of INRs on the generation of time series data.

**Interpretable Time Series**   Seasonal-trend decomposition is a standard tool in time series analysis. The trend encapsulates the slow time-varying behavior of the signal, while seasonal components capture periodicity. These techniques introduce interpretability in time series, which plays an important role in downstream tasks such as forecasting and anomaly detection. The classic approaches for decomposition are the widely used STL algorithm (Cleveland et al., 1990), and its variants (Wen et al., 2019; Bandara et al., 2022). Relevant to this work is the recent N-BEATS architecture (Oreshkin et al., 2020), a deep learning-based model for univariate time series forecasting that provides interpretability capabilities. The model explicitly encodes seasonal-trend decomposition into the network by defining separate trend and seasonal blocks, which fit a low degree polynomial and a Fourier series. While N-BEATS provides an interpretable decomposition, its applications do not cover multivariate time series and data synthesis. Desai et al. (2021) recently proposed TimeVAE, combining a VAE architecture with a trend-seasonality decomposition structure to allow for interpretable generation. However no results highlighting the advantages of this capability were demonstrated.

## 3   FORMULATION

In this Section, we describe the iSIREN network architecture for time series representation and interpretable decomposition, and the HyperTime and iHyperTime networks leveraged for prior learning and new data generation.

### 3.1   TIME SERIES REPRESENTATION

We consider a time series signal encoded by a discrete sequence of $N$ observations $\mathbf{y} = (\mathbf{y}_1, ..., \mathbf{y}_N)$ where $\mathbf{y}_i \in \mathbb{R}^m$ is the $m$-dimensional observation at time $t_i$. This time series defines a dataset $\mathcal{D} = \{(t_i, \mathbf{y}_i)\}_{i=1}^N$ of time coordinates $t_i$ associated with observations $\mathbf{y}_i$. We want to find a continuous mapping $f : \mathbb{R} \to \mathbb{R}^m, t \to f(t)$ that parameterizes the discrete time series, so that $f(t_i) = y_i$ for $i = 1...N$. The function $f$ can be approximated by an implicit neural representation (INR) architecture conditioned on the training loss $\mathcal{L} = \sum_i \|y_i - \hat{f}(t_i)\|^2$. The network is composed of fully-connected layers with sine activations (SIREN) as defined by Sitzmann et al. (2020b):

$$\phi_j(\mathbf{x}_i) = \sin(\omega_0 \mathbf{W}_j \mathbf{x}_i + \mathbf{b}_j) \tag{1}$$

where $\phi_j(\cdot)$ corresponds to the $j^{th}$ layer of the network. A general factor $\omega_0$ multiplying the network weights determines the order of magnitude of the frequencies that will be used to encode the signal. Input and output of the INR are of dimensions 1 and $m$, corresponding to the time coordinate $t$ and the prediction $\hat{f}(t)$. After training, the network encodes a continuous representation of the functional relationship $f(t)$ for a single time series.

#### 3.1.1   INTERPRETABLE SIREN

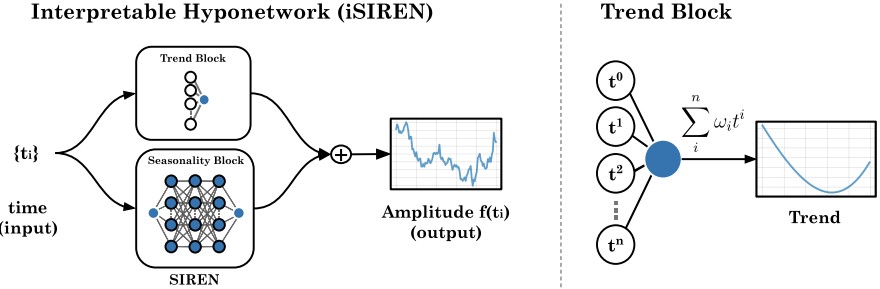

Figure 1: Interpretable SIREN (iSIREN) architecture comprised of two blocks. *Trend Block*:Single FC layer that outputs a polynomial fit of a fixed degree. *Seasonality Block*: SIREN network.

We propose an interpretable architecture to encode time series that reuses the INR with sine activation functions from the previous section. In particular, we assume that our INR follows a classic time series additive decomposition, i.e.,

$$f(t) = f_{\text{tr}}(t) + f_{\text{s}}(t), \tag{2}$$

where $f_{\text{tr}}(t)$ represents the trend component and $f_{\text{s}}(t)$ represents the seasonal component of $f(t)$, respectively. Note that this is a standard assumption for time series decomposition techniques, such as STL and others. We elaborate on the structure of $f_{\text{tr}}(t)$ and $f_{\text{s}}(t)$ in the following.

**Trend block** The trend component of a time series aims to model slow-varying (and occasionally monotonic) behavior. Following the work by Oreshkin et al. (2020) (N-BEATS), for the trend component of the interpretable architecture, we consider a polynomial regressor, i.e.,

$$f_{\text{tr}}(t) = \sum_{i=0}^{p} \mathbf{w}_i^{(\text{tr})} t^i, \tag{3}$$

where $p$ denotes the degree of the polynomial, and $\mathbf{w}_i^{(\text{tr})}$ denotes the weight vector associated with the $i$th degree. In practice, $p$ is chosen to be small (e.g., $p = 2$) to capture low frequency behavior.

**Seasonality block** The seasonal component of the time series $f_{\text{s}}(t)$ aims to capture the periodic behavior of the signal. In classical techniques, a pre-determined and fixed period (or set of periods) determines the resulting extracted seasonal components (e.g., STL supports single seasonal component, whereas MSTL supports multiple). In this work, we propose to model the seasonal component of the time series as a SIREN with $J$ layers (see Section 3.1). Importantly, a single layer SIREN has been shown to be structurally similar to a Fourier mapped perception (FMP). An FMP $h : d_{\text{in}} \to d_{\text{out}}$ is a perceptron with identity activation and Fourier mapped input, i.e., for $\mathbf{x} \in \mathbb{R}^{d_{\text{in}}}$ we have

$$h(\mathbf{x}) = \mathbf{W}^{\text{FS}} \cdot \gamma(\mathbf{x}) + \mathbf{b}^{\text{FS}}, \qquad \gamma(\mathbf{x}) = \begin{bmatrix} \cos(2\pi \mathbf{B} \cdot \mathbf{x}) \\ \sin(2\pi \mathbf{B} \cdot \mathbf{x}) \end{bmatrix}, \tag{4}$$

where $\mathbf{W}^{\text{FS}} \in \mathbb{R}^{d_{\text{out}} \times 2M}$ is a Fourier weight matrix, $\mathbf{b}^{\text{FS}} \in \mathbb{R}^{d_{\text{out}}}$ and $\gamma(\mathbf{x})$ is a Fourier mapping of the input $\mathbf{x}$, with $\mathbf{B} \in \mathbb{R}^{M \times d_{\text{in}}}$ the Fourier mapping matrix of $M$ frequencies. For a particular choice of $\mathbf{B}$ and $\mathbf{b}^{\text{FS}} = \mathbf{0}$, an FMP is equivalent to a Fourier series representation and the Fourier mapping $\gamma(\mathbf{x})$ is equivalent to a single layer SIREN with fixed Fourier mapping matrix (Benbarka et al., 2022). A Fourier series is precisely what is used in the N-BEATS algorithm to model the seasonal component of the time series (Oreshkin et al., 2020). The benefit of using a SIREN is that the Fourier mapping becomes a trainable parameter – enabling the learning of the seasonal component of the time series in an *unsupervised* manner.

**Training of iSIREN** The training of iSIREN is performed in a supervised manner with an MSE reconstruction loss, as described for SIREN in Section 3.1. However, in order to stabilize the training, the process is performed in two stages: 1) we train the Trend Block for 100 epochs, computing the MSE loss between the ground truth time series $\mathbf{y}$ and the output of the block: $\mathcal{L}_1 = \sum_i \|y_i - \hat{f}_{\text{tr}}(t)\|^2$. This leads to a smooth approximation of the time series, which we use as initial guess for the second stage. 2) We then train the Trend and Seasonality blocks together, computing the MSE reconstruction loss between the ground truth and the added output of both iSIREN blocks: $\mathcal{L}_2 = \sum_i \|y_i - \hat{f}(t_i)\|^2$.

## 3.2 TIME SERIES GENERATION WITH HYPERTIME

In Figure 2-top, we display a diagram of the HyperTime architecture, which allows us to leverage INRs to learn priors over the space of time series. The Set Encoder (green network), composed of SIREN layers (Sitzmann et al., 2020b), takes as input a pair of values, corresponding to the time-coordinate $t$ and the time series signal $f(t)$. Each pair of values is thus encoded into an embedding and evaluated by the HyperNet decoder (blue network), composed of fully-connected layers with ReLU activations (MLP). The output of the HyperNet is a one-dimensional vector that contains the network weights of an INR which encodes the time series data from the input. The INR architecture used within HyperTime is our interpretable SIREN (iSIREN), discussed in the previous section and illustrated in Figure 1. However, in Section 4.2 we will also consider the case of using SIREN as target INR. Following previous works (Sitzmann et al., 2020a), in the context of hypernetworks we refer to these predicted INRs as HypoNets in order to avoid ambiguities.

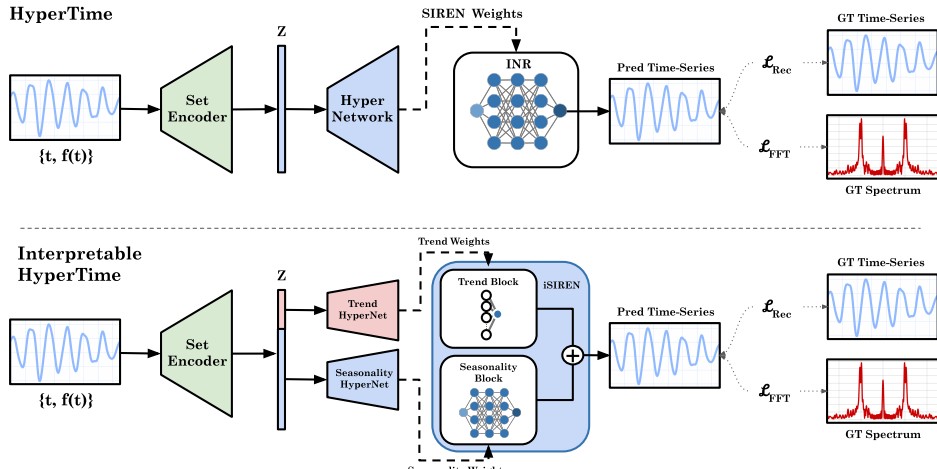

Figure 2: Diagram of HyperTime network architecture. Each pair of time-coordinate $t$ and time series $f(t)$ is encoded as a 40-values embedding $Z$ by the Set Encoder. The HyperNet decoder learns to predict HypoNet weights from the embeddings. During training, the output of the HyperNet is used to build a HypoNet and evaluate it on in the input time-coordinates. The loss is computed as a difference between $f(t)$ and the output of the HypoNet $\hat{f}(t)$.

During the training of HyperTime, we use the weights predicted by the HyperNet decoder to instantiate a HypoNet and evaluate it on the input time-coordinate $t$, to produce the predicted time series value $\hat{f}(t)$. The entire chain of operations is implemented within the same differentiable pipeline, and hence the training loss can be computed as the difference between the ground truth time series signal $f(t)$ and the value predicted by the HypoNet $\hat{f}(t)$.

The use of a Set Encoder in the architecture introduces permutation invariance in the computation. This enables the encoding of data with missing values or irregularly sampled, which are common occurrences in time series datasets. After training, the Set Encoder is able to generate latent embeddings $Z$ to encode entire time series. In Section 4.2, we show that these embeddings can be interpolated to produce new samples from the distribution learned by the network, enabling the synthesis of new time series from known ones.

**Training Loss**     The training of HyperTime is done by optimizing the following loss, which contains an MSE reconstruction term $\mathcal{L}_{\text{rec}}$ and two regularization terms $\mathcal{L}_{\text{weights}}$ and $\mathcal{L}_{\text{latent}}$, for the network weights and the latent embeddings, respectively:

$$\mathcal{L} = \underbrace{\frac{1}{N}\sum_{i=1}^{N}\left\|f(t_i) - \hat{f}(t_i)\right\|^2}_{\mathcal{L}_{\text{rec}}} + \lambda_1 \underbrace{\frac{1}{W}\sum_{j=1}^{W}w_j^2}_{\mathcal{L}_{\text{weights}}} + \lambda_2 \underbrace{\frac{1}{Z}\sum_{k=1}^{Z}z_k^2}_{\mathcal{L}_{\text{latent}}} + \lambda_3 \mathcal{L}_{\text{FFT}}. \tag{5}$$

Since we are working with time series, we want to ensure an accurate reconstruction not only of the signal but also of its spectral composition. Hence, we introduce an additional Fourier-based loss $\mathcal{L}_{\text{FFT}}$ that penalizes differences in the distribution of frequencies of the INR-encoded time series with respect to the ground truth data:

$$\mathcal{L}_{\text{FFT}} = \frac{1}{N}\sum_{i=1}^{N}\left\|\text{FFT}[f(t)]_i - \text{FFT}[\hat{f}(t)]_i\right\|. \tag{6}$$

In Section 4.2, we show that $\mathcal{L}_{\text{FFT}}$ is crucial for the accurate reconstruction of some datasets that show large variations of amplitudes in the frequency domain.

## 3.3    INTERPRETABLE HYPERTIME

In the previous section, we proposed a hypernetwork architecture (HyperTime) to learn a prior over a dataset of time series, encoding each sample as a small embedding $Z$. Although the HT architecture

uses interpretable SIRENs as HypoNetworks, the decomposition between trend and seasonality signals is limited to the predicted INR, and is not explicitly reflected in the generated embeddings.

In Figure 2-bottom, we display our interpretable HyperTime (iHT) network, which builds over the previous HyperTime architecture to incorporate interpretability into the latent representation. Here the embedding $Z$ is split in two parts $Z_T$ and $Z_S$, corresponding to the encodings of the Trend and Seasonality signals. Each of these two embeddings is decoded by a different HyperNetwork (Trend HyperNet and Seasonality HyperNet) and affects the weights of a different block (Trend Block and Seasonality Block) within the predicted iSIREN. Only at the output of the iSIREN both signals are recomposed into a predicted time series, which is compared against the ground truth signal via the reconstruction and spectral losses ($\mathcal{L}_{Rec}$ and $\mathcal{L}_{FFT}$).

The training of iHyperTime is performed in two stages, in order to improve stability, as explained for iSIREN in Section 3.1.1. After training, the Set Encoder produces a decomposed encoding of time series into interpretable embeddings $Z_T$ and $Z_S$. As with the HyperTime architecture, these embeddings can be interpolated to produce new unseen time series. However, as we will show in Section 4.2, we can now interpolate only one of the embeddings while leaving the other fixed. This way we can interpolate between time series with different trends while keeping the seasonality fixed and vice-versa allowing us to introduce fixed temporal structures to the data generation which can be used to inject domain expertise in cases where there is not sufficient data.

## 4 EXPERIMENTS

### 4.1 RECONSTRUCTION

We start by analyzing the reconstruction performance of our model iSIREN over univariate and multivariate time series datasets from the UCR archive (Bagnall et al., 2017). We selected datasets with varying lengths and numbers of features, and sampled 100 time series from each of them, training a single INR for each time series. We compare against other INRs with different activations functions (Tanh, ReLU, Sine) and with ReLU positional encoding (P.E.) (Mildenhall et al., 2020) using INRs with equal size (3 hidden layer of 60 FC neurons). In addition, iSIREN presents 4 more trainable parameters per time series feature, due to the Trend block.

In Figure 3, we display two measures of reconstruction error averaged over 100 samples of a dataset (FordA). Figure 3-top displays the MSE training loss of each model in log space as a function of the training epoch. We observe that both models based on sine activations converge to similarly low values of error. We can also see that iSIREN's convergence is delayed during the first training iterations due to the sequential training of the Trend and Seasonality blocks, detailed in Section 3.1.1.

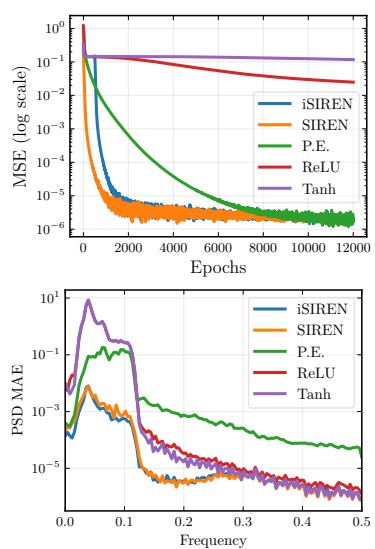

Figure 3: Comparison of iSIREN with other INR encodings. *Top:* MSE loss. *Bottom:* MAE of power spectral density.

In Figure 3-bottom, we focus on the accurate reconstruction of the GT spectral distribution. We illustrate the MAE of the power spectrum density (log) vs the frequency. Here we observe too that the lowest errors correspond to SIREN and iSIREN for all frequencies, with a slight advantage of our method for low frequencies. The reconstruction across all datasets has been summarized in Table 1, where we observe that iSIREN presents the best reconstruction performance, especially in terms of the spectral distribution. Moreover, for multivariate data iSIREN shows the lowest errors over all metrics and datasets, with a particularly large advantage in the case of PhonemeSpectra.

Finally in Figure 4, we illustrate the capacity of iSIREN to perform an interpretable decomposition of time series. In column (a) we show the signal reconstruction, and in columns (b) and (c) the

corresponding outputs of the Trend and Seasonality blocks. As it would be expected, the Trend curve follows the overall shape of the signal, while the Seasonality prediction oscillates around zero. For a qualitative comparison, in columns (d) and (e) we see good agreement with an analogous decomposition performed by a traditional method (STL (Cleveland et al., 1990)). In contrast with iSIREN, STL requires prior knowledge of the frequency of the signal's seasonality.

Table 1: Comparison using MSE on time space and MAE in frequency space (FFT) of implicit networks using different activation functions and of iSIREN on univariate and multivariate datasets.

| Dataset | iSIREN (Ours) | | SIREN | | P.E. | | ReLU | | Tanh | |
|---|---|---|---|---|---|---|---|---|---|---|
| | FFT | MSE | FFT | MSE | FFT | MSE | FFT | MSE | FFT | MSE |
| *Univariate* | | | | | | | | | | |
| Crop | 1.4e-3 | 5.6e-6 | 1.4e-3 | 1.6e-6 | **6.8e-4** | **7.3e-7** | 5.4e-1 | 2.1e-2 | 8.6e-1 | 6.0e-2 |
| Energy | **4.1e-3** | **5.3e-6** | 1.8e-2 | 1.2e-5 | 1.3e-1 | 7.7e-4 | 1.5e+0 | 4.9e-2 | 1.9e+0 | 8.3e-2 |
| FordA | **1.7e-2** | **4.9e-6** | 1.9e-2 | 6.2e-6 | 3.1e-1 | 2.1e-3 | 2.5e+0 | 1.3e-1 | 2.8e+0 | 1.4e-1 |
| NonInv | **3.6e-2** | **1.2e-5** | 4.0e-2 | 1.3e-5 | 1.1e-1 | 1.3e-4 | 1.0e+0 | 2.2e-2 | 1.3e+0 | 4.6e-2 |
| Phalanges | **1.4e-3** | **2.1e-6** | 3.8e-3 | 1.8e-6 | 7.6e-3 | 1.2e-5 | 2.4e-1 | 3.8e-3 | 7.5e-1 | 8.4e-2 |
| Stock | **2.5e-3** | 5.1e-6 | 4.4e-3 | **1.4e-6** | 4.3e-2 | 1.2e-4 | 6.2e-1 | 1.2e-2 | 8.9e-1 | 3.8e-2 |
| *Multivariate* | | | | | | | | | | |
| Cricket | **3.9e-1** | **4.1e-4** | 4.5e-1 | 4.2e-4 | 1.7e+0 | 3.7e-3 | 3.5e+0 | 1.7e-2 | 3.9e+0 | 3.1e-2 |
| MotorImagery | **5.1e+0** | **2.1e-3** | 7.2e+0 | 6.2e-3 | 1.1e+1 | 2.4e-2 | 1.0e+1 | 2.6e-2 | 1.1e+1 | 3.0e-2 |
| PhonemeSpectra | **2.9e-2** | **2.1e-6** | 4.2e-1 | 2.7e-4 | 1.8e+0 | 5.9e-3 | 3.0e+0 | 1.5e-2 | 3.4e+0 | 2.0e-2 |

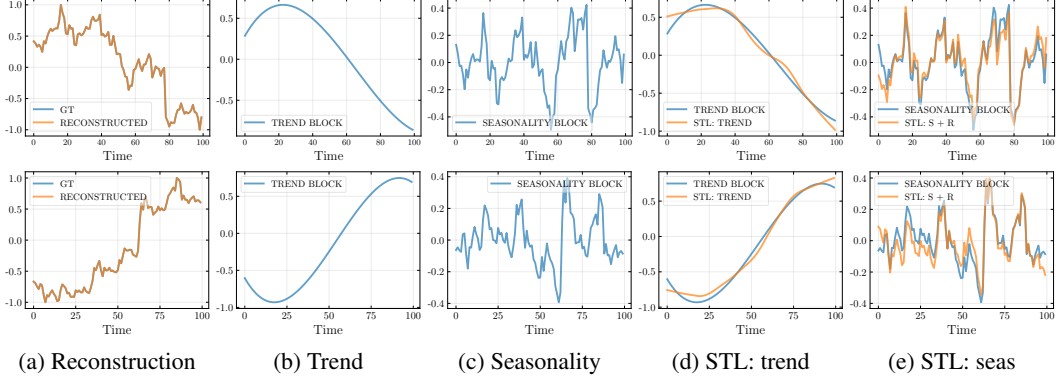

(a) Reconstruction  (b) Trend  (c) Seasonality  (d) STL: trend  (e) STL: seas

Figure 4: Interpretable decomposition of iSIREN on two time series from the `Stock` dataset. (a) Ground truth and iSIREN reconstruction. (b) Trend Block output. (c) Seasonality Block output. Columns (d) and (e) compare the output of iSIREN blocks with classic STL decomposition.

## 4.2 TIME SERIES GENERATION

We leverage our proposed hypernetwork architectures for time series generation via the interpolation of latent representations. We follow the experimental set up proposed by Alaa et al. (2021) to evaluate our model on multiple datasets in terms of two metrics: 1) Predictive score (MAE), which measures how well the synthetic samples can be used to train a model and predict on real samples in a forecasting task. 2) F1-score, measuring the quality of the generated data via precision and recall metrics averaged over all time steps Sajjadi et al. (2018).

In Figure 5, we show qualitative evaluations of the generation performance for iHyperTime, and compare against multiple methods, including RealNVP Dinh et al. (2017) and current state-of-the-art for time series generation (TimeGAN Yoon et al. (2019b) and Fourier Flows Alaa et al. (2021)). We see a good agreement of iHyperTime for all datasets, while other methods fail to match the original data in some cases. Performance results using the Predictive score and F1-score metrics are summarized in Table 2. In addition to iHyperTime, we evaluate the following 3 other variations of the architecture:

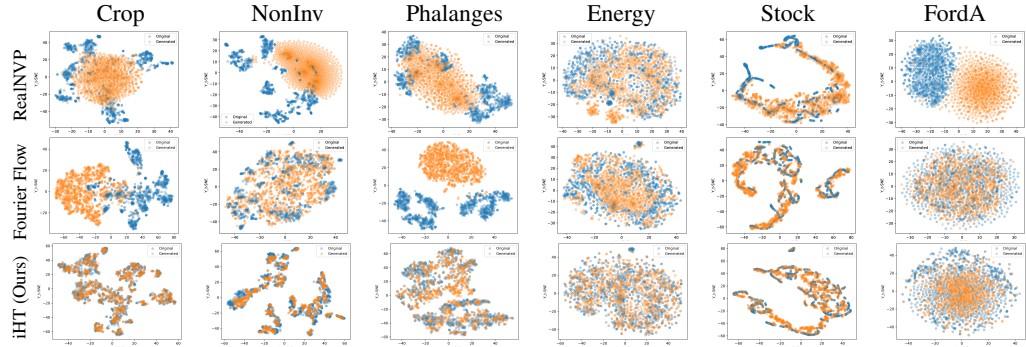

Figure 5: t-SNE visualization of real (blue) and synthetic (orange) data for all univariate datasets (in columns), using different time series generation methods (in rows).

**HyperTime (no FFT):** this architecture is directly based on one of the original hypernetwork architectures from SIREN (Sitzmann et al.).

**HyperTime (w/FFT):** same as previous, but including the FFT loss in the training.

**HyperTime (iSIREN):** same as previous, but using iSIREN as the predicted hyponetwork, thus generating time-series representations with built-in interpretable TS decomposition.

Table 2: Top: Performance scores for data generation using baselines and iHyperTime. Bottom: HyperTime ablations (in blue: values that improve over the baselines).

|  | Crop | NonInv | Phalan. | Energy | Stock | FordA |
|---|---|---|---|---|---|---|
| **RealNVP** | | | | | | |
| *MAE* | 0.170 | 0.038 | 0.073 | 0.036 | 0.019 | 0.115 |
| *F1 Score* | 0.981 | 0.986 | 0.976 | 0.964 | 0.977 | **0.999** |
| **TimeGAN** | | | | | | |
| *MAE* | 0.048 | – | 0.108 | 0.056 | 0.173 | – |
| *F1 Score* | 0.831 | – | 0.960 | 0.479 | 0.938 | – |
| **Fourier Flows** | | | | | | |
| *MAE* | 0.040 | 0.018 | 0.056 | **0.030** | **0.010** | 0.024 |
| *F1 Score* | 0.991 | 0.990 | 0.992 | 0.936 | 0.990 | 0.998 |
| **iHT (Ours)** | | | | | | |
| *MAE* | **0.039** | **0.004** | **0.024** | 0.056 | 0.011 | **0.009** |
| *F1 Score* | **0.999** | **0.997** | **0.997** | **0.997** | **0.995** | 0.996 |
| *Ablations* | | | | | | |
| **HT (no FFT)** | | | | | | |
| *MAE* | 0.040 | 0.005 | 0.023 | 0.058 | 0.012 | 0.170 |
| *F1 Score* | 0.999 | 0.996 | 0.996 | 0.998 | 0.995 | 0.084 |
| **HT (w/ FFT)** | | | | | | |
| *MAE* | 0.040 | 0.005 | 0.023 | 0.057 | 0.013 | 0.007 |
| *F1 Score* | 0.999 | 0.997 | 0.999 | 0.997 | 0.994 | 0.998 |
| **HT (iSiren)** | | | | | | |
| *MAE* | 0.039 | 0.004 | 0.024 | 0.057 | 0.013 | 0.008 |
| *F1 Score* | 0.999 | 0.997 | 0.999 | 0.997 | 0.995 | 0.997 |

All hypernetwork architectures consistently generate high-quality synthetic time series, outperforming the baselines across most datasets and metrics. This indicates that the data generated by the HT and iHT architectures presents a high predictive utility for model training (Predictive score) and a large overlap with the distribution of the real data (F1 score).

An interesting failure case is presented with the generation of `FordA` samples by the **HT (no FFT)** model (in red). Here both evaluation metrics indicate a low performance. As shown in the supplemental material, the unusual diversity of spectral distributions present in the `FordA` dataset suggests that HT is unable to learn an effective reduced representation due to the complexity of the data. However,

the issue is solved by the introduction of the spectral loss $\mathcal{L}_{FFT}$, as observed on the metrics for the **HT (w/FFT)** model. Moreover, the inclusion of this loss does not result in a loss of performance over other datasets.

Finally, in Figure 6 we illustrate the interpolation of interpretable embeddings from iHT. This model produces independent encodings $Z_T$ and $Z_S$ for the trend and seasonality signals, enabling additional control over the synthesis process by interpolating them separately. Figure 6a shows an interpolation of $Z_S$ from source to target, leaving $Z_T$ unchanged. By the end of the process, the interpolated time series has acquired the seasonality of the target signal, but maintains its original trend. Conversely, in Figure 6b we show an interpolation of the trend $Z_T$ while fixing the seasonality $Z_S$. In both cases, we observe a smooth transition of the signal corresponding to the interpolated embedding, indicating a good generalization of the iHT architecture.

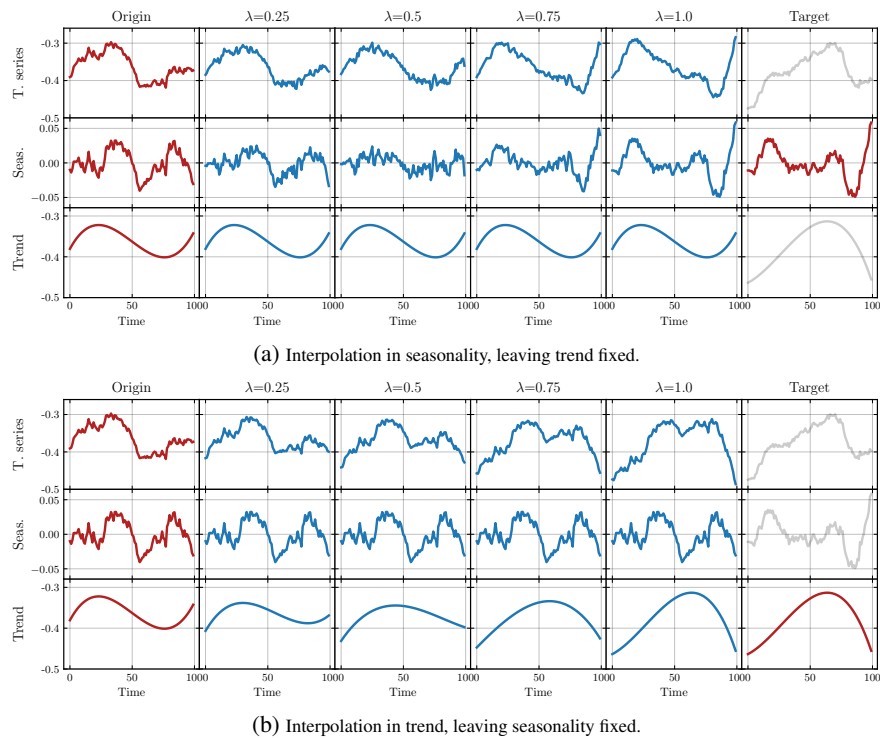

(a) Interpolation in seasonality, leaving trend fixed.

(b) Interpolation in trend, leaving seasonality fixed.

Figure 6: (a) Interpolation of seasonality, with fixed trend. (b) Interpolation in trend, with fixed seasonality. In red: original TS (1st column) and target seasonality/trend (last column).

## 5 CONCLUSIONS

In this work, we explored the use of INRs for the encoding and analysis of both univariate and multivariate time series data. We proposed a novel INR architecture (iSIREN) that enables learning an interpretable (seasonality-trend) continuous representation of time series data. We showed that our model outperforms other INRs in terms of the accurate reconstruction of the signal and its spectral distribution, with particular high performance for multivariate time series. Following this, we presented HyperTime, a hypernetwork architecture that leverages iSIRENs to learn a prior over a time series dataset. We demonstrated that by interpolating between embeddings learned by HT, we can generate new synthetic time series. To guide the training of HT, we introduced a Fourier-based loss that enforces the accurate reconstruction of the spectral distribution of the ground truth signals.

Finally we introduced iHyperTime, a hypernetwork model that incorporates interpretability into its latent representation of time series. We compared both HT and iHT against state-of-the-art methods for time series generation, showing improved results across most datasets and generative evaluation metrics. Furthermore, we showed that the disentangled seasonal-trend embeddings learned by iHT can be leveraged to introduce constraints into the generation, with potential applications for the incorporation of expert knowledge in the synthesis of new time series.

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
