# OpenReview forum: "HyperTime: Implicit Neural Representations for Time Series Generation"
_ICLR.cc/2023/Conference — Submitted to ICLR 2023_

### Official Review · Reviewer_Wscx · 2022-10-16

**Confidence:** 3
**Correctness:** 3
**Technical Novelty And Significance:** 2
**Empirical Novelty And Significance:** 2
**Recommendation:** 5

**Clarity, Quality, Novelty And Reproducibility:**

I find the paper to be exceptionally clear, and the quality of experiments to be relatively high. However, I believe that the novelty of the method is very limited - existing tools from a method were applied to a slight variation of a previous signal type, and it was not compared against state-of-the-art coordinate-based network architectures for signal representation, rather against baselines which have shown to completely fail. The method seems reproducible.

**Strength And Weaknesses:**

In my opinion, the strengths of the paper are:
- The paper tackles an important problem which is improving the quality of neural signal representations of 1D signals, and generalization across instances of these signals.
- The paper is written very clearly, and the method is explained well. The figures are very detailed and describe the proposed architecture concisely.

I think that the main weakness of the paper is novelty. Specifically:
- The iSIREN architecture proposed is a SIREN, but with a factored out low-frequency component. This has already been proposed in [1] (where the output of a coordinate-based network is the sum of two different functions), although a slightly different function implementation is used here for the trend component instead of another low-frequency biased SIREN.
- The generalization aspect, HyperTime, effectively just uses the SIREN + hypernetwork setup, with the same set-encoder used in the SIREN paper, for generalization. This generalization scheme has already been shown to work for coordinate-based representations of images, so it is not surprising it could also work for 1D signals, considering that the original SIREN paper also fits 1D signals in the form of audio.
- It seems that the results don’t outperform the standard SIREN by much? Figure 3 shows that SIREN and iSIREN seem to converge similarly, and Table 1 shows that they are much more similar to each other than to any of the other methods. It is difficult to evaluate the relative magnitude of the difference in these numbers.
- Additionally, I don’t find the baseline comparisons to be fair. There exist other methods for coordinate-based networks (ex: hybrid implict-explicit architectures, SAPE, etc…) which can be used for signal representation and generalization. Simply comparing versus other nonlinearities in the network architecture is not a fair comparison as the SIREN paper already has shown that they simply don’t work for high-frequency signal memorization, so they are not valid baselines to compare to an improvement on SIREN.
- The generalization visualization in Figure 5 shows that the synthetic and real data distributions match, but the real data distribution doesn’t seem to be clustered. Thus I wonder if the prior learned by this hypernetwork would allow the useful aspects of generalization, such as solving inverse problems for example? In the original SIREN paper, this is used to perform inpainting of images - is something similar done here for time series?


[1] Geometry-Consistent Neural Shape Representation with Implicit Displacement Fields. Yifan, Wang and Rahmann, Lukas and Sorkine-hornung, Olga. ICLR, 2021.


**Summary Of The Paper:**

This paper proposes to use coordinate-based networks for time-series data representation. This is done by proposing a new architecture which separates a time series into a trend (low-frequency) and seasonal (high-frequency) component by representing each independently with a set of weights and a SIREN, and then summing them. The method proposes to generalize over these coordinate-based representations of time series data using hypernetwork architectures: one which outputs standard SIREN representations of time series, and one which outputs the modified iSIREN architecture proposed with the decomposition of signal frequencies. The paper demonstrates that this new architecture results in improved time series fitting, and that the hypernetwork architecture places similar time series signals at similar places in the latent space.

**Summary Of The Review:**

The paper proposes a method for representing time series data with a coordinate-based network architecture based on SIRENs, and generalizing across these representations with hypernetworks. Overall, I find the method to be described and evaluated well. However, the architecture proposed is heavily based on previous work, and is not compared to other state-of-the-art coordinate-based representation architectures which have emerged for other applications. The generalization aspect is interesting, but is not novel as it is just a drop-in of the method used in SIREN, and is not shown to be useful for a useful application in 1D signal processing, such as solving inverse problems.

POST REBUTTAL UPDATE: After reading the other reviews and responses, I am not inclined to change my score. I now understand that the purpose of iSIREN is not to improve directly on SIREN, but instead to provide an interpretable breakdown. However, the argument of using two separate SIRENs (as in [1]) requiring an additional hyperparameter is not convincing, as iSIREN also needs to choose the parametric form for one of the networks. While I believe the FFT loss is novel in this context (although it has been used before in imaging applications), I find that the paper still combines a lot of known methods into a new type of data. This is interesting and important for this application domain, but doesn't propose anything new or enable significantly more interpretable function fitting. For this reason, I view the paper as borderline for acceptance.

---

> ### Author Response · Authors · 2022-11-19
> **Response to Reviewer Wscx (1/2)**
>
> Thank you for taking the time to review our paper, and for providing constructive feedback.
>
> > The iSIREN architecture proposed is a SIREN, but with a factored out low-frequency component. This has already been proposed in [1] (where the output of a coordinate-based network is the sum of two different functions), although a slightly different function implementation is used here for the trend component instead of another low-frequency biased SIREN. [1] Geometry-Consistent Neural Shape Representation with Implicit Displacement Fields. Yifan, Wang and Rahmann, Lukas and Sorkine-hornung, Olga. ICLR, 2021.
>
> Thank you for directing us to the paper, we will include it in the related work as INRs for signal decomposition. In our work, we propose as the trend component (low-frequency) a low degree polynomial, as this is how the trend is traditionally modelled in time-series decomposition (e.g. STL, N-BEATS). From [1], the base surface is modelled with a low-frequency component using a SIREN with a low omega  ($\omega_B=15$) and the displacement is modelled using a SIREN with high omega ($\omega_D=60$). As explained in section B.2, for very smooth surfaces, the $\omega_B=15$ is already sufficient to represent the ground truth surface and the displacement has little impact, so in order to create a frequency separation, the w has to be reduced. This adds an additional hyper-parameter to the problem, as one would need to tune the relative values between the two omegas to ensure separation.
>
> > The generalization aspect, HyperTime, effectively just uses the SIREN + hypernetwork setup, with the same set-encoder used in the SIREN paper, for generalization. This generalization scheme has already been shown to work for coordinate-based representations of images, so it is not surprising it could also work for 1D signals, considering that the original SIREN paper also fits 1D signals in the form of audio.
>
> Although our initial architecture is similar to the hypernetwork by Sitzmann et al., we introduce multiple differences:
> * We use iSIREN as hyponetwork, instead of SIREN.
> * We introduce a novel FFT loss that is critical to guide the training for some datasets (in Table 2 we show that the original SIREN+hypernetwork fails to generalize without this loss).
> * We introduce an interpretable TS decomposition of the latent embeddings (see Figure 2-bottom).
> * We leverage these architectures for time-series generation via interpolation of embeddings. SIREN+hypernetwork are used for image inpainting.
>
> For a more in-depth discussion of differences with SIREN, please see the corresponding subsection in the general discussion of novelty. In order to clarify these differences in the manuscript, we have added an itemized description of the architectures in Section 4.2.
>
> > It seems that the results don’t outperform the standard SIREN by much? Figure 3 shows that SIREN and iSIREN seem to converge similarly, and Table 1 shows that they are much more similar to each other than to any of the other methods. It is difficult to evaluate the relative magnitude of the difference in these numbers.
>
> The reconstruction of time-series with SIREN is usually very accurate. The main purpose of iSIREN is to provide an interpretable decomposition of the data, without losing significant accuracy with respect to SIREN. We found, however, that iSIREN outperformed SIREN in many cases, especially for multivariate data (Table 1), and for the low frequencies components (Figure 3).
>
> > Additionally, I don’t find the baseline comparisons to be fair. There exist other methods for coordinate-based networks (ex: hybrid implicit-explicit architectures, SAPE, etc…) which can be used for signal representation and generalization. Simply comparing versus other nonlinearities in the network architecture is not a fair comparison as the SIREN paper already has shown that they simply don’t work for high-frequency signal memorization, so they are not valid baselines to compare to an improvement on SIREN.
>
> The main purpose of our paper is to develop a novel method for time-series representation and generation, with the introduction of interpretable elements in both tasks. iSIREN introduces interpretability in the representation of time-series, and does so without losing accuracy with respect to SIREN. Accuracy is already very high in this case, due to the relative simplicity of the signals.
>
> The generation of high-quality time-series data is a much more complex task. In Figure 5 and Table 2 we perform a comparison against current state-of-the-art methods for time-series generation, using standard quality and fidelity metrics.

---

> > ### Author Response · Authors · 2022-11-19
> > **Response to Reviewer Wscx (2/2)**
> >
> > > The generalization visualization in Figure 5 shows that the synthetic and real data distributions match, but the real data distribution doesn’t seem to be clustered. Thus I wonder if the prior learned by this hypernetwork would allow the useful aspects of generalization, such as solving inverse problems for example? In the original SIREN paper, this is used to perform inpainting of images - is something similar done here for time series?
> >
> > Figure 5 shows t-SNE visualizations applied directly to the synthetic and real time-series data (not to their latent embedding representations). This is part of the standard evaluation for time-series generation methods, and seeks to establish a match between real and synthetic data distributions (together with the quantitative metrics from Table 2). Examples of this analysis can be found in state-of-the-art methods for time-series generation, such as Fourier Flows and TimeGAN.
> >
> > There are multiple potential applications for time-series, such as forecasting and inpainting (in time-series this is known as imputation). Here we have focused on the task of time-series generation, which arguably is one of the most important applications within medicine and finance, and one that is still relatively unexplored. However, we plan to investigate other applications as future work.
> >
> > > I find the paper to be exceptionally clear, and the quality of experiments to be relatively high. However, I believe that the novelty of the method is very limited - existing tools from a method were applied to a slight variation of a previous signal type, and it was not compared against state-of-the-art coordinate-based network architectures for signal representation, rather against baselines which have shown to completely fail. The method seems reproducible.
> >
> > We did our best to give a clear description of our work, thank you for noticing our effort.
> >
> > Regarding novelty: we wrote a general discussion of the novelty of the paper. We dedicate a subsection of the discussion specifically to emphasize the differences between our method and the architectures from SIREN, both for representation and for generation. For the generation task, these difference are (in short): 1) iSIREN hyponetwork. 2) Novel and critical FFT loss. 3) Interpretable TS decomposition of embeddings+hypernetworks. 4) Different application (generation instead of inpainting).
> >
> > Regarding evaluation: we compared against state-of-the-arts methods for time-series generation, showing improved results for a large majority of datasets and generation metrics. For representation, our purpose was to provide an interpretable representation of time-series, ideally without loss of accuracy with respect to SIREN, which already has a very high accuracy for this type of data. We were glad to find that in some cases iSIREN presented higher accuracy, but this was not the intended purpose of our representation. Please see the general discussion of results for a more detailed explanation.

---

> > > ### Comment · Reviewer_Wscx · 2022-12-02
> > > **Response**
> > >
> > > Thank you for the detailed response, I have updated my review. I don't have any additional questions.

---

> > > > ### Author Response · Authors · 2022-12-09
> > > > **Post Rebuttal Reply**
> > > >
> > > > Dear reviewer Wscx,
> > > >
> > > > Thank you for taking the time to read our responses. Regarding iSIREN, we would like to clarify that our proposed trend block does not require the tuning of an additional hyperparameter: for all experiments we used a fixed low-degree polynomial following the literature (N-BEATS).
> > > >
> > > > Regarding novelty: thank you for considering our "general discussion on novelty".
> > > > We would like to emphasize that the key motivation for our innovations is the generation of synthetic time series data (done with iHyperTime). Time series generation is an important problem which has not previously been addressed using INRs, and in which our method clearly outperforms all current state-of-the-art methods (e.g. Fourier Flows), with the additional benefit of incorporating some degree of interpretability into the generation.

---

### Official Review · Reviewer_H1Gq · 2022-10-24

**Confidence:** 4
**Clarity, Quality, Novelty And Reproducibility:** The work is clear and incrementally o…
**Correctness:** 4
**Technical Novelty And Significance:** 3
**Empirical Novelty And Significance:** 2
**Recommendation:** 6

**Strength And Weaknesses:**

Strengths:
1. I think that enforcing accuracy in the Fourier domain adds powerful constraints that should generate better results.
Weaknesses:
1. The f(t) = ftr(t) + fs(t) decomposition introduces a very strong assumption: What happens with non seasonal signals?
2. Even though the results are good and generally better, the experiments are not conclusive in stating the superiority of the proposed pipeline. This is important because the method is not based on fundamental principles but on common sense ad-hoc decisions. Thus, it is the experiments that should state whether the pipeline is better or not.

**Summary Of The Paper:**

The authors present the iSIREN pipeline, designed to accurately reconstruct signals. To do it, they use latent representations of the data set. They also work with metrics in the FFT domain to enforce an even better reconstruction.

**Summary Of The Review:**

As stated above, the work is incrementally novel. Even though very good results are obtained, it is difficult to assess whether they are general (not many experiments) or conclusive (marginal improvements).

---

> ### Author Response · Authors · 2022-11-19
> **Response to Reviewer H1Gq**
>
> We thank the reviewer for the positive feedback. We address the concerns below.
>
> > The f(t) = ftr(t) + fs(t) decomposition introduces a very strong assumption: What happens with non seasonal signals?
>
> What we refer as seasonality is in fact the higher frequencies of the signal that do not correspond to the trend. This includes the periodic components (proper seasonality) and the non-periodic ones (residuals).
>
> > Even though the results are good and generally better, the experiments are not conclusive in stating the superiority of the proposed pipeline. This is important because the method is not based on fundamental principles but on common sense ad-hoc decisions. Thus, it is the experiments that should state whether the pipeline is better or not.
>
> Although our method for time-series generation greatly surpassed the baselines across the large majority of datasets and metrics, we believe that this was not clearly conveyed in Table 2, due to the large number of variations of our architecture that are present in the table (we apologize for the lack of clarity). We modified the table in the manuscript in order to improve clarity. Below is an abbreviated version of the table showing only iHyperTime.
>
> |                   |          | Crop      | NonInv    | Phalan.   | Energy    | Stock     | FordA     |
> | :--------------- | :------- | :-------: | :-------: | :-------: | :-------: | :-------: | :-------: |
> | **RealNVP**       |      MAE | 0.170     | 0.038     | 0.073     | 0.036     | 0.019     | 0.115     |
> |                   | F1 Score | 0.981     | 0.986     | 0.976     | 0.964     | 0.977     | **0.999** |
> | **TimeGAN**       |      MAE | 0.048     | --        | 0.108     | 0.056     | 0.173     | --        |
> |                   | F1 Score | 0.831     | --        | 0.960     | 0.479     | 0.938     | --        |
> | **Fourier Flows** |      MAE | 0.040     | 0.018     | 0.056     | **0.030** | **0.010** | 0.024     |
> |                   | F1 Score | 0.991     | 0.990     | 0.992     | 0.936     | 0.990     | 0.998     |
> | **iHT (Ours)**    |      MAE | **0.039** | **0.004** | **0.024** | 0.056     | 0.011     | **0.009** |
> |                   | F1 Score | **0.999** | **0.997** | **0.997** | **0.997** | **0.995** | 0.996     |
>
> The new version of Table 2 compares our iHyperTime architecture (iHT) against all baselines, showing a clear advantage (in bold) for most datasets and metrics. The table still includes the other three HyperTime architectures at the bottom, but labels them as ablations of iHyperTime. They represent simpler versions of iHyperTime, with increasing modifications of the architecture, such as the addition of the FFT loss, and the use of iSIREN hyponetworks. Our iHyperTime model incorporates all of these modifications, and adds a Trend-Seasonality decomposition of the latent embeddings and hypernetworks, enabling the interpretable generation of time-series as shown in Figure 6 and in Section 5 of the supplemental material. Despite this added interpretable capability, iHyperTime greatly outperforms state-of-the-art methods for time-series generation. Ablation models also outperform baselines, and this is shown in blue in the new table.
>
> Furthermore, we have added in Section 5 of the supplemental material, a new expanded version of Table 2 that includes training and inference times for all baselines and hypernetwork architectures. Our method is faster at training than TimeGAN and Fourier Flows, which are the current state-of-the-arts methods for time-series generation. At inference, Fourier Flows is faster than our models in most cases, although for long time-series (NonInv) inference time grows considerably larger.
>
> > The work is clear and incrementally original.
>
> We wrote a general discussion of the novelty of the paper at the beginning of this rebuttal. We dedicate a subsection of the discussion specifically to emphasize the differences between our method and the architectures from SIREN, both for representation and for generation. For the generation task, these difference are (in short): 1) iSIREN hyponetwork. 2) Novel and critical FFT loss. 3) Interpretable TS decomposition of embeddings+hypernetworks. 4) Different application (generation instead of inpainting). Moreover, we are the first to introduce interpretable components into the time series generation process.

---

### Official Review · Reviewer_jVNg · 2022-10-26

**Confidence:** 4
**Correctness:** 3
**Technical Novelty And Significance:** 3
**Empirical Novelty And Significance:** 3
**Recommendation:** 6

**Clarity, Quality, Novelty And Reproducibility:**

Clarity and quality is good, while novelty is fairly ok. Reproducibility is hard because of missing details, though the underlying models, e.g., SIREN and hypernet are easy to implement.

**Strength And Weaknesses:**

* Strength

(i) The work is well written and the motivation of using implicit neural network is appreciated because of lack of the similar work on this direction.

(ii) By separating the trend component from the seasonality, the model has better interpretability. Besides, it also offers a way for trend/seasonality decomposition.

(iii) Hypernet is an interesting angle here, as it saves time from training a new model for incoming data.

* Weakness

(i) The work is largely built-on SIREN and the migrating to time-series data is not challenging, given the flexibility of implicit network itself.

(ii) While the decomposition of tend and seasonality is interesting, it is hard to justify they are the components that are claimed to be, esp. for the case of seasonality, i.e., SIREN can model seasonality (more broadly, periodic pattern) well, but you may not claim that what SIREN has modelled is truly seasonality.

**Summary Of The Paper:**

In this work, the authors propose an implicit neural representations method for time-series data, namely iSIREN. The key different from the original SIREN is that iSIREN has modelled the trend and seasonality explicitly. Then authors show that the parameters of iSIREN can be produced by a hypernetwork (HyperTime or iHyperTime), such that the new generative model can be produced on-the-fly without any training by feeding the time-series data into hypernetwork. The proposed method is demonstrated for reconstruction and generation tasks cross many datasets and surpassing some other baselines.

**Summary Of The Review:**

Overall it is a nice paper with a good demonstration of implicit network for time-series data. I may upgrade the score if the seasonality modelling part is further elaborated.

---

> ### Author Response · Authors · 2022-11-19
> **Response to Reviewer jVNg**
>
> Thank you for taking the time to review our paper, and for providing constructive feedback.
>
> > The work is largely built-on SIREN and the migrating to time-series data is not challenging, given the flexibility of implicit network itself.
>
> The reconstruction of time-series data is indeed not challenging, given the flexibility of implicit networks. However, this is only the starting point for a series of contributions that include interpretable decomposition, spectrum-aware training via the novel FFT loss, and the interpretable generation of time-series data, which enables the introduction of expert knowledge into the process (see general discussion of novelty). In particular, we focus our main analysis in the comparison against state-of-the-art methods for time-series generation, showing improved results.
>
> > While the decomposition of trend and seasonality is interesting, it is hard to justify they are the components that are claimed to be, esp. for the case of seasonality, i.e., SIREN can model seasonality (more broadly, periodic pattern) well, but you may not claim that what SIREN has modelled is truly seasonality.
>
> This is correct. The trend is modelled with a low degree polynomial, as done in STL decomposition and in N-BEATS by Oreshkin et al. What we refer as seasonality is in fact the higher frequencies of the signal that do not correspond to the trend. This includes the periodic components (proper seasonality from STL) and the non-periodic ones (residuals). We will include a clarification of this point in the manuscript.
>
> > Reproducibility is hard because of missing details, though the underlying models, e.g., SIREN and hypernet are easy to implement.
>
> We have now included a link to the full code of the paper.
> We have also included new versions of Tables 1 and 2 in supplemental material, with additional details such as standard deviations and training/inference times. Moreover, we have updated sections 4.3 and 5.2 of the supplemental material with additional implementation details.

---

> > ### Comment · Reviewer_jVNg · 2022-12-12
> > **Post-rebuttal**
> >
> > While I still have some doubts on the technical contribution, I'd like to increase my score because the author made the code available.

---

> > > ### Author Response · Authors · 2022-12-12
> > > **Post-rebuttal reply**
> > >
> > > Dear reviewer jVNg,
> > >
> > > thank you for taking our rebuttal into consideration and for raising your score. We appreciate your feedback on our work.

---

### Official Review · Reviewer_wjaa · 2022-10-28

**Confidence:** 4
**Correctness:** 3
**Technical Novelty And Significance:** 2
**Empirical Novelty And Significance:** 2
**Recommendation:** 3

**Clarity, Quality, Novelty And Reproducibility:**

The clarity and quality of writing needs to be significantly improved. Most parts of the provided formulas are not properly introduced and defined, which makes it very hard to read. For instance almost none of the variables in eq. (5) are defined anywhere in the paper.
The novelty is very marginal, as both SIREN and hypernetworks are already established methods as well as using a time series additive decomposition into trend and seasonal components are well established in standard time series analysis.

The results are not reproducible, as no code is provided. Moreover, no standard deviations are provided for different initializations of the models. Hence, the results are not reproducible.

**Strength And Weaknesses:**

Using a series decomposition for SIREN and SIREN-based hypernetworks in novel up to my knowledge. However, using an additive time series decomposition into trend and seasonal component is one of the most standard approaches in time series analysis. The paper provides many and versatile experiments showing a strong performance of the provided models on many different tasks.

However, there are open questions and weaknesses that need to be addressed:

* Is the better performance of iSIREN due to the additive decomposition technique or simply because it uses slightly more parameters than standard SIREN?
* What are the weaknesses of the provided models, e.g. slower training/inference time and so on?
* Adding an FFT penalty term to the loss function seems to be a great idea. Is it novel or has it been used in this context already before?
* No standard deviations are provided. It is not clear if the obtained results are robust.

**Summary Of The Paper:**

This paper proposes, based on SIREN, a new interpretable INR architecture for time series called iSIREN. Building up on that, the paper proposes to use the iSIREN set-up in the context of hypernetworks for time series generating. Both methods differ from its plain counter version (SIREN and SIREN-based hypernetwork) in that it splits the model into a classic time
time series additive decomposition, i.e. trend and seasonal component. The paper provides a variety of different experiments showing its superior performance compared to baseline models.

**Summary Of The Review:**

The paper introduces iSIREN and a hypernetwork based on iSIREN, which consists of SIREN together with a time series additive decomposition. The novelty is very marginal and the results are not outstanding. Moreover, the results are not reproducible. Finally, major parts of the paper need to be rewritten. Hence I recommend to reject the manuscript in its current form.

---

> ### Author Response · Authors · 2022-11-19
> **Response to Reviewer wjaa (1/2)**
>
> We thank the reviewer for the detailed review. We address the concerns below.
>
> > using an additive time series decomposition into trend and seasonal component is one of the most standard approaches in time series analysis.
>
> We have addressed concerns about novelty in the general section "trend-seasonality decomposition".
>
> > Is the better performance of iSIREN due to the additive decomposition technique or simply because it uses slightly more parameters than standard SIREN?
>
> The number of trainable parameters added by the trend block is very small relative to the total size of the network. For reference, in univariate time-series the seasonality block has 7440 trainable parameters, and the trend block contains only 4 trainable parameters. The addition of a small number of weights is not enough to justify the improvements in performance observed in Table 1. We believe that the addition of an inductive bias in the trend block (low degree polynomial) helps guide the proper reconstruction of the signal, especially in terms of avoiding spurious high-frequency components in the spectral distribution.
>
> > What are the weaknesses of the provided models, e.g. slower training/inference time and so on?
>
> The main weakness of our model lies in the seasonality block. What we refer as seasonality is in fact the higher frequencies of the signal that do not correspond to the trend. This includes the periodic components (proper seasonality) and the non-periodic ones (residuals).
>  We will include a clarification of this on the final manuscript.
>
> Regarding training/inference times: we have added in Section 5 of the supplemental material, a new expanded version of Table 2 that includes training and inference times for all baselines and hypernetwork architectures. We include below an abbreviated version of the table containing only the training and inference times. Our method is faster at training than TimeGAN and Fourier Flows, which are the current state-of-the-arts methods for time-series generation. At inference, Fourier Flows is faster than our models in most cases, although for long time-series (NonInv) inference time grows considerably larger.
>
> |  |  | *Crop* | *NonInv* | *Phalan.* | *Energy* | *Stock* | *FordA* |
> |----------------------|----------|:----------:|:-----------:|:----------:|:---------:|:---------:|:---------:|
> | **RealNVP**      | Time (training) | 51.9 | 100.3 | 19.6 | 242.6 | 46.3 |73.1|
> |       | Time (generation) |  0.03  |   0.20   |   0.06    |   0.07   |  0.06   |  0.09   |
> | **TimeGAN**          | Time (training) | 1799 | -- | 3177 | 3942 | 3872 |--|
> |       | Time (generation) |  0.6   |    --    |    0.6    |   0.7    |   0.7   |   --    |
> | **Fourier Flows**  | Time (training) | 864.1 | 543.7 | 75.2 | 1652.1 | 172.6 |607.2|
> |       | Time (generation) |  0.07  |   1.46   |   0.08    |  0.087   |  0.09   |   1.2   |
> | **iHT (Ours)**           | Time (training) | 208.5 | 146.3 | 71.4 | 443.5 | 98.2 |205.3|
> |       | Time (generation) |  0.5   |   0.4    |    0.3    |   2.1    |   0.7   |   0.5   |

---

> > ### Author Response · Authors · 2022-11-19
> > **Response to Reviewer wjaa (2/2)**
> >
> > > Adding an FFT penalty term to the loss function seems to be a great idea. Is it novel or has it been used in this context already before?
> >
> > Thank you for acknowledging the technical significance of the FFT loss. We developed it specifically for this problem, based on the importance of an accurate reconstruction of the signal's spectrum in the context of time-series analysis. However, we believe that it has potential applications for other data sources and network models. We have only found one work that uses a similar penalty term in the context of image super-resolution [1].
> >
> > [1] Fourier Space Losses for Efficient Perceptual Image Super-Resolution. Dario Fuoli, Luc Van Gool, Radu Timofte. ICCV, 20
> >
> > > No standard deviations are provided. It is not clear if the obtained results are robust.
> >
> > We apologize for not including the standard deviations. We have added new versions of Tables 1 and 2 in the supplemental material, that include the standard deviations for all combinations of models and time-series datasets.
> >
> > > The novelty is very marginal, as both SIREN and hypernetworks are already established methods as well as using a time series additive decomposition into trend and seasonal components are well established in standard time series analysis.
> >
> > Our work leverages SIREN and hypernetworks to provide new methods for time-series representation and generation, adding interpretable components into both tasks. For representation, we combine SIREN with a polynomial fitting to produce a Trend-Seasonality decomposed encoding. The fact that trend-seasonality is a standard decomposition for time-series is an advantage in terms of interpretability by a human observer, since practitioners are used to interact with data via this kind of decomposition.
> >
> > For time-series generation, we leverage hypernetworks but we introduce multiple important modifications to the architecture and training process:
> > * We introduce interpretability in the hyponetwork, by using iSIREN instead of SIREN.
> > * We introduce a novel FFT loss that is critical to guide the training for some datasets (in Table 2 we show that the original SIREN+hypernetwork fails to generalize without this loss).
> >  * We introduce a second level of interpretability by doing TS decomposition of the latent embeddings and hypernetworks (see Figure 2-bottom).
> >  * We leverage these architectures for time-series generation via interpolation of embeddings.
> >
> > As a result, we provide a new method for time-series generation that outperforms current state-of-the-art techniques. Moreover, we are the first to introduce interpretability into the process of time-series generation (via the TS decomposition of embeddings).
> >
> > For a more in-depth discussion of differences with SIREN, please see the corresponding subsection in the general discussion of novelty.
> >
> > > The results are not reproducible, as no code is provided.
> >
> >  As promised in Section 4.4 of the supplemental material, we are uploading the full code as part of the rebuttal. This follows the indications of the official guidelines, to avoid making the code immediately public. We apologize for not submitting the code earlier, due to IP compliance requirements. We will make the code public after the review process.

---

### Author Response · Authors · 2022-11-19
**General Response to All Reviewers**

We thank all the reviewers for their valuable feedback, which has helped to improve the quality of our paper. We were happy to see that there was good agreement about some of the strengths of our work:
* The relevance of the problem (jVNg, Wscx).
* The clarity of exposition (jVNg, Wscx).
* The technical novelty of the FFT loss (wjaa, H1Gq).
* The quality of the experiments (wjaa, Wscx).

We also found common topics on the perceived weaknesses of our work: Novelty or similarity with previous hypernetwork architectures, Results, Trend-Seasonality Decomposition and Reproducibility. We consider these common topics in a general discussion. We also address each comment from the reviewers individually.

The new additions and modifications to the manuscript are summarized as follows:

* We rewrite the introduction to improve the clarity of the contributions.
* We rewrite Section 4.2 and update Table 2 in order to improve clarity of the results and to highlight the strong performance achieved by iHyperTime.
* We include an extended version of Table 1 in Supplementary material, which includes the standard deviations.
* We include an extended version of Table 2 in Supplementary material, which includes the standard deviations across 5 random seeds. It also includes training and inference times for all models.
* We rewrite the reproducibility and implementation details in Supplementary material to improve clarity and to include further details.
* We submit a link to the source code and pre-trained models.

We hope that our answers and the revision of the paper will address your concerns.

---

> ### Author Response · Authors · 2022-11-19
> **General discussion on novelty**
>
> We are grateful for the reviewers' feedback, which helped us realize that the novelty of the paper was not sufficiently clear. We have modified the introduction to provide a more concise summary of the contributions.
>
> ### Contributions
> In this paper, we propose a novel methodology that utilizes INRs to encode and generate time series data based on interpretable latent representations. To the best of our knowledge, we are the first to incorporate an interpretable decomposition into the generation of time-series. Our contributions are as follows:
> **Representation and Generation of time-series using INRs:** We introduce iSIREN, an INR architecture for multivariate time-series representation which provides an interpretable trend-seasonality decomposition of the data. We show that interpretability does not lead to a loss of reconstruction accuracy, and in some cases increases the spectral reconstruction quality. Moreover, we leverage a hypernetwork for time-series generation via interpolation of learned embeddings.
>  **Spectral Loss:** To improve the training of the hypernetwork, we introduce a novel spectral loss that enforces the correct reconstruction of the signal's spectral distribution. We show that for some datasets this loss plays a crucial role in the learning process.
> **Interpretable time-series generation:** We propose iHyperTime, a hypernetwork architecture for time-series generation that learns a disentangled seasonal-trend representation of time series, enabling the introduction of expert knowledge into the synthesis process. We compare iHyperTime against current state-of-the-art methods for time-series generation, showing improved results in terms of standard fidelity metrics.
>
>  ### Differences with SIREN
>
> Some reviews state that our architectures for reconstruction and generation are too similar to SIREN, or even the same. Below we emphasize the differences of our models, both for reconstruction and generation:
> * For reconstruction, iSIREN implements an interpretable decomposition of the univariate/multivariate signal by adding a low degree polynomial as inductive bias.
> * For generation, we evaluate 4 different hypernetwork architectures:
> 	1. HyperTime (no FFT): this architecture is directly based on one of the original hypernetwork architectures from SIREN for image inpainting (Sitzmann et al.). However, we show that this architecture does not generalize well for some datasets (FordA, Table 2).
> 	2.  HyperTime (w/FFT): same architecture as (1), but we include the FFT loss in the training, which solves the learning issues such as with FordA.
> 	3. HyperTime (iSIREN): same as (2), but we have replaced the hyponetwork by an iSIREN network, thus generating time-series representations with built-in interpretable TS decomposition.
> 	4. iHyperTime: this is our final architecture. It combines all of the previous additions (FFT loss, iSIREN hyponetwork), and provides a TS decomposition of the embedding, enabling the insertion of expert knowledge into the generation process, as shown in Figure 6. The full architecture for iHyperTime is illustrated in Figure 2-bottom.
>
> Hence, our Interpretable HyperTime architecture differs from SIREN in the inclusion of an iSIREN hyponetwork, the TS decomposition of the embedding (and their respective hypernetworks), and the use of a novel and critical FFT loss. In order to improve the clarity of these contributions, we have added an itemized description of the architectures in Section 4.2 of the paper.

---

> > ### Author Response · Authors · 2022-11-19
> > **General discussion on trend-seasonality decomposition**
> >
> > ## TREND-SEASONALITY DECOMPOSITION
> > Some reviews remarked that additive TS decomposition is not novel for time-series. This decomposition is, indeed, a standard method for time-series analysis, as stated in Section 2. We see this is an advantage of our representation in terms of interpretability, since practitioners within time-series analysis are used to interact with data via this kind of decomposition.
> >
> > It is also worth mentioning that TS decomposition plays two different roles in our method:
> > 1.  In iSIREN it provides a built-in interpretable representation of the time-series signal. This comes without loss of reconstruction accuracy, and in many cases the introduction of an inductive bias provides a noticeable improvement of the quality of the reconstruction, as shown in Figure 3.
> >  2. In iHyperTime, the TS decomposition is applied to the latent embeddings (and the hypernetwork blocks), adding interpretability into the generation process, by allowing the independent manipulation of trend and seasonality components, as shown in Figure 6.
> >  Interpretability is a desirable property on many time-series tasks, and there are multiple works on TS decomposition in time-series forecasting tasks (e.g. N-BEATS). However, despite the critical importance of time-series generation in areas such as medicine and finance, there are no previous works incorporating interpretability into the generation process (as mentioned in the related work section, the recently proposed TimeVAE combines TS decomposition with a VAE architecture, with potential applications for interpretable generation, but without results demonstrating this capability).

---

> > ### Author Response · Authors · 2022-11-19
> > **Discussion of results**
> >
> > Some reviews remarked that our results were not conclusive on the advantage of our proposed methods over established techniques.   We would like to re-instate some of our results in a more concise way.
> >
> > The main purpose of our work is to develop a novel method for time-series representation and generation, with the introduction of interpretable elements in both tasks. The iSIREN model introduces interpretability in the representation of time-series, and does so without loss of accuracy with respect to SIREN, which has already been shown to provide a very accurate reconstruction of a wide variety of signals, including audio (i.e. univariate time-series). In Table 1, we showed that our model presents a higher accuracy than SIREN in many cases, although this was not the main purpose of our representation. In Figure 4, we illustrated the trend-seasonality decomposition of our method, and compared with the traditional STL decomposition. The full set of decomposition results can be found in Section 4 of the supplemental material.
> >
> > For time-series generation, we developed multiple architectures based on hypernetworks and compared very favorably against current SOTA methods for time-series generation. For the evaluation (Table 2) we used standard fidelity metrics, which are specific for the task of time-series generation, and have been widely used in similar works (Fourier Flows, TimeGAN). Although our method for time-series generation greatly surpassed the baselines across the large majority of datasets and metrics, we believe that this was not clearly conveyed in Table 2, due to the large number of variations of our architecture that are present in the table (we apologize for the lack of clarity). We modified the table in the manuscript in order to improve clarity.
> >
> > |                   |          | Crop      | NonInv    | Phalan.   | Energy    | Stock     | FordA     |
> > | :--------------- | :------- | :-------: | :-------: | :-------: | :-------: | :-------: | :-------: |
> > | **RealNVP**       |      MAE | 0.170     | 0.038     | 0.073     | 0.036     | 0.019     | 0.115     |
> > |                   | F1 Score | 0.981     | 0.986     | 0.976     | 0.964     | 0.977     | **0.999** |
> > | **TimeGAN**       |      MAE | 0.048     | --        | 0.108     | 0.056     | 0.173     | --        |
> > |                   | F1 Score | 0.831     | --        | 0.960     | 0.479     | 0.938     | --        |
> > | **Fourier Flows** |      MAE | 0.040     | 0.018     | 0.056     | **0.030** | **0.010** | 0.024     |
> > |                   | F1 Score | 0.991     | 0.990     | 0.992     | 0.936     | 0.990     | 0.998     |
> > | **iHT (Ours)**    |      MAE | **0.039** | **0.004** | **0.024** | 0.056     | 0.011     | **0.009** |
> > |                   | F1 Score | **0.999** | **0.997** | **0.997** | **0.997** | **0.995** | 0.996     |
> > |                   |          |           |           |           |           |           |           |
> > | _ABLATIONS_       |          |           |           |           |           |           |           |
> > |                   |          |           |           |           |           |           |           |
> > | **HT (no FFT)**   |      MAE | [0.040]()   | [0.005]()   | [0.023]()   | 0.058     | 0.012     | 0.170     |
> > |                   | F1 Score | [0.999]()   | [0.996]()   | [0.996]()   | [0.998]()   | [0.995]()   | 0.084     |
> > | **HT (w/ FFT)**   |      MAE | [0.040]()   | [0.005]()   | [0.023]()   | 0.057     | 0.013     | [0.007]()   |
> > |                   | F1 Score | [0.999]()   | [0.997]()   | [0.999]()   | [0.997]()   | [0.994]()   | 0.998     |
> > | **HT (iSiren)**   |      MAE | [0.039]()   | [0.004]()   | [0.024]()   | 0.057     | 0.013     | [0.008]()   |
> > |                   | F1 Score | [0.999]()   | [0.997]()   | [0.999]()   | [0.997]()   | [0.995]()   | 0.997     |
> >
> > The new version of Table 2 compares our iHyperTime architecture (iHT) against all baselines, showing a clear advantage (in bold) for most datasets and metrics. The table still includes the other three HyperTime architectures at the bottom, but labels them as ablations of iHyperTime. They represent simpler versions of iHyperTime, with increasing modifications of the architecture, such as the addition of the FFT loss, and the use of iSIREN hyponetworks. Our iHyperTime model incorporates all of these modifications, and adds a Trend-Seasonality decomposition of the latent embeddings and hypernetworks, enabling the interpretable generation of time-series as shown in Figure 6 and in Section 5 of the supplemental material. Despite this added interpretable capability, iHyperTime greatly outperforms state-of-the-art methods for time-series generation. Ablation models also outperform baselines, and this is shown in blue in the new table. The exception to this is the bad performance over the FordA dataset of the HT (no FFT) model, which is based on the original hypernetwork architecture by Sitzmann et al. and is discussed in our paper.

---

### Decision · Program_Chairs · 2023-01-20

**Decision:**

Reject

**Justification For Why Not Higher Score:**

I provided a details meta-review in Part 1.

**Justification For Why Not Lower Score:**

N/A

**Metareview: Summary, Strengths And Weaknesses:**

The paper combines a series of known methods namely SIREN and Hypernets with the separation of a time series into a trend and seasonal component proposes enhanced interpretability and performance. A new

Strength:
- The presentation of the paper is great.
- The FFT loss component helps in generation tasks.
- code was not provided, but the authors made it available which is great

Weaknesses:
- The reviewers collectively question the novelty of the paper as it is heavily built on top of known methods. This is not necessarily a weakness. However, when we put this in perspective with the performance gains presented especially in the time-series modeling task, the results are indeed incremental and not of substantial use.
- In modeling (representing) time series, there are numerous advanced baselines that are overlooked in the current work. Models such as Legendre Memory Units [A Voelker et al. NeurIPS 2021], Hippo [Gu et al. NeurIPS 2020], Lipschitz RNNs [Erichson et al. ICLR 2021], Long Expressive Memory [Rusch et al. ICLR 2022], Closed-form Continuous-time Neural nets [Hasani et al. Nature MI 2022], and many more.
- The FFT loss in one generation task helps with achieving better performance. However, its use in time series modeling tasks seems to be redundant. Some of the baseline methods presented outperform iSIREN with a normal MSE loss. The paper fails to demonstrate how this loss could be of value in the "representation" tasks. Furthermore, the introduction of the FFT loss is not novel. It has been used in other spaces before.

During the discussion period, the authors provided a comprehensive explanation of their work and tried to address the reviewers' concerns. However, after this period most reviewers are voting the paper to be of borderline nature.

I believe the paper in its current form is not ready for publication and could benefit from extensive experimental results with up-to-date baselines and benchmarks.




**Summary Of Ac-Reviewer Meeting:**

N/A